# From Biomarkers to the Molecular Mechanism of Preeclampsia—A Comprehensive Literature Review

**DOI:** 10.3390/ijms241713252

**Published:** 2023-08-26

**Authors:** Magda Rybak-Krzyszkowska, Jakub Staniczek, Adrianna Kondracka, Joanna Bogusławska, Sebastian Kwiatkowski, Tomasz Góra, Michał Strus, Wojciech Górczewski

**Affiliations:** 1Department of Obstetrics and Perinatology, University Hospital, 30-688 Krakow, Poland; mistrus@su.krakow.pl; 2Department of Gynecology, Obstetrics and Gynecological Oncology, Medical University of Silesia, 40-211 Katowice, Poland; jstaniczek@sum.edu.pl; 3Department of Obstetrics and Pathology of Pregnancy, Medical University of Lublin, 20-081 Lublin, Poland; adriannakondracka@wp.pl; 4Department of Biochemistry and Molecular Biology, Centre of Postgraduate Medical Education, 01-813 Warsaw, Poland; joanna.boguslawksa@cmkp.edu.pl; 5Department Obstetrics and Gynecology, Pomeranian Medical University, 70-111 Szczecin, Poland; kwiatkowskiseba@gmail.com; 6Clinical Department of Gynecology and Obstetrics, Municipal Hospital, John Paul II in Rzeszów, 35-241 Rzeszów, Poland; minddin@gmail.com; 7Independent Public Health Care Facility “Bl. Marta Wiecka County Hospital”, 32-700 Bochnia, Poland; woj-tekg1_9@op.pl

**Keywords:** preeclampsia, biomarkers, molecular biology, predictive value of tests, maternal–fetal health

## Abstract

Preeclampsia (PE) is a prevalent obstetric illness affecting pregnant women worldwide. This comprehensive literature review aims to examine the role of biomarkers and understand the molecular mechanisms underlying PE. The review encompasses studies on biomarkers for predicting, diagnosing, and monitoring PE, focusing on their molecular mechanisms in maternal blood or urine samples. Past research has advanced our understanding of PE pathogenesis, but the etiology remains unclear. Biomarkers such as PlGF, sFlt-1, PP-13, and PAPP-A have shown promise in risk classification and preventive measures, although challenges exist, including low detection rates and discrepancies in predicting different PE subtypes. Future perspectives highlight the importance of larger prospective studies to explore predictive biomarkers and their molecular mechanisms, improving screening efficacy and distinguishing between early-onset and late-onset PE. Biomarker assessments offer reliable and cost-effective screening methods for early detection, prognosis, and monitoring of PE. Early identification of high-risk women enables timely intervention, preventing adverse outcomes. Further research is needed to validate and optimize biomarker models for accurate prediction and diagnosis, ultimately improving maternal and fetal health outcomes.

## 1. Introduction

Preeclampsia (PE) is one of the most common obstetric illnesses during pregnancy [1], affecting around 3–8% of pregnant women worldwide, making it a leading cause of gestational-related morbidity and mortality [2]. The proteinuria and the new-onset hypertension after 20 weeks of gestation are the hallmark symptoms of PE [3]. However, PE might affect multiple organ systems (respiratory, hepatic, urinary, neuroendocrine, and circulatory) leading to the fetal growth restriction, preterm birth, and other adverse fetal outcomes. These symptoms accompanying hypertension allow the recognition of PE according to the criteria of numerous gynecological and obstetrician societies [4,5]. Although the clinical manifestations of PE usually do not appear before week 20 of gestation, the molecular pathways leading to its onset are believed to occur relatively early in pregnancy [6]. At present, the etiology of disease and thus the effective prediction of preeclampsia before it strikes are still not fully determined. The pieces of evidence indicate that the complexity of pathophysiology and etiology of PE varies between early and late onset of preeclamptic cases. The early-onset type, known as preterm preeclampsia developing before week of 34 of gestation, is recognized as a reason of the defect of placentation, whilst maternal etiology dominates in the occurrence of the late onset of preeclampsia (i.e., term PE occurring after week 34 of gestation) [7]. Due to its multifaceted etiology and pathogenesis, it is impossible to find a single good candidate to predict the occurrence of preeclampsia. Therefore, a constellation of biomarkers including biochemical, ultrasound, and physiologic maternal features such as BMI, age or the smoking status are considered to foresee the probability of the risk of occurrence of PE [8]. 

The Fetal Medicine Foundation (FMF) created a first-trimester screening paradigm that integrates maternal variables with biochemical indicators (PlGF and PAPP-A) and biophysical markers (uterine artery pulsatility index, UtA-PI, and MAP). PE screening for this subtype is therapeutically beneficial since this model predicts preterm PE at 75% with a 10% false-positive rate (FPR). However, the detection rate for predicting term PE is only 41%, and at present, any biochemical markers do not improve the predictive value of the algorithm of prediction of PE based on the USG markers and maternal risk factors [9]. Therefore, the search for the new biochemical as well as molecular predictors of preeclampsia is still a “hot” topic of research investigations. At present, some biochemical markers such as soluble FMS-like tyrosine kinase 1 (sFIT-1), PP-13, GDF15, ADAM12 [10] as well as molecular markers, e.g., cell free ribonucleic acid circulated in maternal blood [11], are considered as candidates for non-invasive tests to increase the efficacy of risk classification for PE pregnancies. Perhaps in the future, these biomarkers will allow the implementation by the physicians of preventive measures for high-risk women. 

This review aims to synthesize and summarize the current state of knowledge regarding the biomarkers allowing the prediction of the late onset of PE. The review specifically focuses on biochemical markers tested in maternal blood or urine and provides an in-depth analysis of their molecular mechanism for the diagnosis of PE.

## 2. Pathogenesis of PE

Sufficient blood flow to the placenta is essential for a correct outcome of pregnancy. During normal implantation, the placental trophoblasts (of fetal origin) invade the uterus inducing the remodeling of uterine spiral arteries, making them wide and low resistant once. This provides adequate placental perfusion to nourish the growing fetus. 

It is believed that the pathogenesis of PE is marked by the defective and inadequate trophoblast invasion into maternal decidua and its artery. The incomplete spiral artery remodeling leads to improper placental perfusion. The placental cells living under prolonged starvation and a hypoxia environment start to secret into maternal bloodstream numerous factors including sFlt1 or soluble endoglin (sENG) influencing the extensive maternal endothelial dysfunction [3,12] and supporting the maternal immune response, the generation of oxidative stress as well as the activation of the maternal coagulation system. All of these functions support the development of hypertension, proteinuria, and very often lead to failure of organs other than kidneys (Figure 1) [13,14,15].

PE may be classified as early-onset and late-onset depending on the timing, pathophysiology, and clinical implications. 

### 2.1. Early-Onset PE

Preeclampsia detected at or after 34 weeks of pregnancy is referred to as being of an early onset type [16]. The condition may be categorized as early-onset preeclampsia, requiring delivery prior to 34 weeks of gestation based on time. This type of preeclampsia is linked to the insufficient trophoblast invasion as a reason of strong inflammatory maternal reaction (Figure 1) [3,13,15]. It is known that immunologic abnormalities observed as an altered profile of lymphocytes T helper (TH) and an elevation level of the CD19+CD5+ B lymphocyte support the PE phenotype. The CD19+CD5+ B lymphocytes are the major cause of forming polyreactive antibodies, especially angiotensin II type 1 receptor (AT1R) autoantibodies [17]. The incorrect gestation switch of lymphocyte T helper from the Th2 to the Th1 subpopulation support inflammation as Th1 cells secrete numerous pro-inflammatory cytokines, including interleukin-12 and interleukin-18. Moreover, the changes in the profile of natural killer cells (NK), as well as their communication with the placental cells, differ between the preeclamptic and normotensive cases. Natural killer cells are one of the subpopulation of uterine cells being localized at the maternal–fetal interface, and they are involved in early placental development, specifically trophoblast invasion and remodeling of the spiral arteries [18,19,20]. Some studies demonstrate that the communication between NK immunoglobulin-like receptor (KIR) and human leukocyte antigen (HLA)-C presented on the trophoblast is failure [19]. Moreover, the deficiency of the maternal CD56+/NKp46+ and CD56+bright/NKp46+ cells is the hallmark of preeclampsia; indeed, the depletion in the levels of both fractions of NK occurs three/four months before the onset of the disease [19,21,22]. 

### 2.2. Late-Onset PE

Late-onset PE may be induced due to maternal genetic predisposition to cardiovascular and metabolic diseases [12]. The diseased placenta releases factors causing widespread damage to the endothelium of the maternal organs such as kidneys, brain, and liver. Pathologic analysis of the adrenal glands and liver has shown infarction, necrosis, and hemorrhage. The kidneys may reveal the presence of severe glomerular endotheliosis, and the heart may show endocardial necrosis. Podocyturia and a reduced glomerular filtration rate (by 40%) have been observed in women with PE. The autopsy findings of women who died from eclampsia also revealed cerebral edema and intracerebral parenchymal hemorrhage. Research reveals that women with PE showed impaired endothelium-dependent vasorelaxation along with a subtle rise in pulse pressure and blood pressure before the onset of hypertension and proteinuria (Figure 1) [13,14,15]. Severe PE may also become an underlying factor for the appearance of the HELLP (hemolysis, elevated liver enzymes, low platelets) syndrome, eclampsia (seizures), and/or restricted fetal growth (Figure 1) [23].

## 3. Biomarkers and Their Molecular Pathway Development and Prediction of PE

The elucidation of the pathophysiology of preeclampsia has led to the development of various assays that estimate the maternal concentrations of biochemical markers (angiogenic or anti-angiogenic factors); these assays may further aid in administration of improved diagnosis [7].

Evidence suggests the occurrence of an imbalance amongst pro-angiogenic factors (Vascular endothelial growth factor (VEGF), platelet growth factor (PlGF), and transforming growth factor-β (TGF-β)) and anti-angiogenic factors (Soluble FMS-like tyrosine kinase-1 (sFlt-1) and Soluble Endoglin (sEng)) which contribute to the pathophysiological effects observed in PE [13,24].

Research suggests that differences in plasma concentration of pro-angiogenic (TGF-β and PlGF) and anti-angiogenic (sFlt-1 and sEng) factors are associated with PE. Serum samples collected at the time of delivery revealed a significant rise in sFlt-1 concentration and reduced PlGF concentration in PE compared to the controls. These disproportionate levels of anti-angiogenic factors (sFlt-1 and sEng) and pro-angiogenic factors (PlGF, VEGF, and TGF-β) cause maternal endothelial dysfunctions, further leading to the development of renal endotheliosis, hypertension, and blood coagulation [3,13,24].

The most promising screening method for an early diagnosis and prognosis of PE during the third trimester of gestation is to estimate a combination of biomarkers such as PlGF, sFLT1, and sEng with improved sensitivity and specificity [3].

### 3.1. Pregnancy-Associated Plasma Protein-A (PAPP-A)

PAPP-A (pappalysin-1) is a high-molecular-weight (200 KDa, 1547 amino acids) glycoprotein synthesized by the placental trophoblasts and secreted into the maternal bloodstream. PAPP-A interacts with insulin-like growth factors and is significant for the growth of the placenta and fetus. PAPP-A is a metalloproteinase (containing a Zn^2+^-binding site) that is involved in the proteolytic cleavage of the insulin-like growth factor binding protein (IGFBP), eventually regulating the local insulin-like growth factor (IGF) action, which acts as a growth-promoting enzyme essential for fetal and placental development. Furthermore, PAPP-A is also involved in rapid and rigorously controlled growth and development processes such as bone remodeling and peak bone mass accrual (during puberty), folliculogenesis, wound healing, and atherosclerosis [24,25,26].

PAPP-A is a fetoplacental-specific molecule that is being used as a biomarker for predicting PE [25]. In pregnant women, PAPP-A predominately circulates as a PAPP-A/proMBP heterotetrametric (99% (proMBP: proform eosinophil major basic protein)). The inhibition of the proteolytic activity of PAPP-A by proMBP serves to prevent a significant increase in IGFBP-4 proteinase activity. However, a local increase in IGFBP-4 proteinase activity is significant for the development of the placenta. The concentration of the PAPP-A protein is lower in the first trimester and gradually increases throughout the gestation period with concentrations increasing by 100-fold during the first trimester and 10,000-fold during the third trimester compared to the levels observed in non-pregnant women. After delivery, the PAPP-A concentration rapidly drops to basal values [25].

PAPP-A is a biochemical marker used earlier for the screening of chromosomal abnormalities and fetal Down syndrome. Placental pathology can decrease the PAPP-A concentrations. Studies have reported that a decreased concentration of PAPP-A in the first trimester may be associated with abnormal placentation or placental dysfunction and the development of subsequent PE [26,27]. The study conducted by Luewan et al. revealed that pregnancies with reduced PAPP-A concentrations were significantly associated with an increased risk of early-onset preeclampsia. Furthermore, they also confirmed that reduced PAPP-A concentrations, at a cut-off of < 10th percentile, may be used to predict preeclampsia (with 26.1% sensitivity and 9.2% false-positive rate) [26,27]. It was also observed that during the early second trimester of pregnant women developing PE, the PAPP-A concentration may decrease to one third of the concentration compared to the values observed in women without PE. However, the diagnostic utility of PAPP-A concentration during the early second trimester has not been demonstrated. Interestingly, the level of the PAPP-A protein increased with the course of preeclamptic gestation obtaining the highest concentration in the third trimester of pregnancy; indeed, mild and severe PE cases demonstrated a 1.5-fold increase in PAPP-A concentration in comparison to the values observed in healthy pregnancy [25].

At present, it is recognized that the determination of PAPP-A concentration along with other biochemical markers, maternal factors and Doppler ultrasound may be used as an early marker for the screening of PE [26,27]. A combination of screening methods such as Doppler PI, PAPP-A, inhibin A, and PlGF showed a detection rate of 100% for early-onset PE; however, for PE (in general), the detection rate was only 40% with a false-positivity rate of 10% [25].

### 3.2. Placental Growth Factor (PlGF) and Vascular Endothelial Growth Factor (VEGF)

PlGF and VEGF are effective pro-angiogenic factors secreted by the trophoblast cells. PlGF is a glycosylated dimeric protein playing a significant role in placental angiogenesis during early pregnancy, and inducing the growth, differentiation, and invasion of trophoblasts into the maternal decidua [28]. The circulating PlGF concentration is prominently increased during pregnancy; in the first trimester, the concentration of this factor is low and it becomes elevated from the 11th to 12th week onwards, reaching a peak at the 30th week of gestation, after which it declines [29]. PlGF belongs to the VEGF family, and it is primarily expressed in the placenta; however, the small concentrations appear also in several other tissues, such as the heart, skeletal muscles, lungs, liver, bone, and thyroid [29]. VEGF plays a key role in the maintenance of endothelial cell function, specifically the fenestrated endothelium (found in the brain, liver, and glomeruli—the major organs affected by PE). Both PlGF and VEGF are linked by anti-angiogenic factor sFlt-1, secreted endogenously. sFlt-1 selectively binds to PlGF and VEGF, thereby inhibiting the binding of PlGF and VEGF with its membrane receptor [3,25], (Figure 2). This influences the levels of free PlGF and VEGF factors circulating in maternal bloodstream. Moreover, the level of both particles is regulated by the partial pressure of oxygen in the environment [30]. Under hypoxic environment, the cells, including those from placenta, start to activate the hypoxia-inducible factor-1 (HIF-1) [31,32,33]. This transcription factor is responsible for the activation of gene expressions coding for proteins implicated in the process of angiogenesis, including PlGF, VEGF, as well as their receptor, i.e., Flt-1 [34]. Interestingly, although the gene coding for PlGF is under the control of HIF-1α, its level presents a negative correlation to the level of HIF-1α, and this phenomenon seems to be dependent on the type of cells [31]. In placental cells, the level of PlGF is downregulated, but VEGF is upregulated under hypoxic conditions [35,36]. 

This might explain why, under preeclamptic condition strongly related to the low level of oxygen (i.e. at about 2%O_2_), the PlGF concentration is depleted. This might also explain why the total level of VEGF is elevated in preeclampsia. However, the free fraction of this particle is significantly depleted in preeclamptic maternal bloodstream, as a consequence of binding of this particle to its soluble receptor, i.e., sFlt-1 [37].

Studies have indicated that PlGF concentrations can be used for an early diagnosis of PE with a detection rate of 90% and a fixed false positive rate of 5% [3]. The increased sFlt-1 concentration in PE is significantly correlated with disease severity. Both PlGF and sFtl-1 concentrations may diagnose PE at the end of the first trimester of gestation [3,25,38]. Additionally, the ratio of sFlt-1 to PlGF seems a good predictive factor of PE; PE women demonstrate a significant elevation of the sFlt-1/PlGF ratio than the healthy controls [25,39]. Moreover, Rana et al. also suggested that the sFlt-1/PlGF ratio higher than ≥85 might be a marker of the early onset of PE and predicted adverse maternal and fetal outcomes. The sFlt-1/PlGF ratio of ≤38 in women at 24–37 weeks of gestation can be a reliable measure to diagnose the absence of PE (negative predictive value—99.9%) [3,38]

Thus, PlGF is a key molecule in the prediction and diagnosis of PE. However, an estimation of a combination of PlGF, VEGF, and sFlt-1 has shown promising results to trace the changes in the placental vasculature and damage in the endothelium before and during PE. These molecules have been estimated in studies for screening PE [25].

### 3.3. Soluble FMS-Like Tyrosine Kinase-1 (sFlt-1)

sFlt-1 is an antiangiogenic soluble protein that mediates its antagonistic effect by binding to and inhibiting the pro-angiogenic proteins—PlGF and VEGF, thus inducing endothelial dysfunction [3]. sFlt-1 is a splice variant of the membrane bound the FMS-like tyrosine kinase-1 (Flt-1) receptor (also known as VEGFR-1). The sFlt-1 is a 100 kDa protein deficient in the transmembrane and intracellular portion of the active protein Flt [32]. The major source of the sFlt-1 protein in the maternal circulation is the placenta. sFlt-1 is a key factor in the development of PE. The biological actions of VEGF and PlGF would be prohibited by increased sFlt-1 levels, thereby inducing the development of PE. sFlt-1 concentration was found to be extremely elevated in the circulation of women with PE, and this raised concentration existed before the development of hypertension and proteinuria [40]. The serum concentration of sFlt-1 in PE may increase to up to five times higher than the concentrations circulating in normotensive women [24,28,41]. The plasma sFlt-1 concentration increased at 6–10 weeks in women with PE compared to the levels observed in normal pregnancies. Moreover, the plasma sFlt-1 concentrations in early-onset PE and late-onset PE showed elevated levels at the 26th and 29th weeks of gestation, respectively, compared to the levels observed in normal pregnancies [24].

### 3.4. Endothelium-Derived Nitric Oxide (NO)

Endothelium-derived NO is a gaseous molecule acting as a potent vasorelaxant that helps in multiple physiological and pathophysiological functions such as angiogenesis, neovascularization, vessel tone regulation, and regulation of systemic blood pressure. NO acts as a central mediator and modulates the effect of various angiogenic factors (VEGF, PlGF, and TGF-β) to stimulate normal endothelial migration and proliferation. The expression of eNOS is found to be upregulated by the angiogenic factors (TGF-β, VEGF, PlGF) [23,42,43].

In endothelial cells, VEGF binds with its receptor (VEGFR) and activates an endothelial-cell-specific isoform of an NO-producing enzyme endothelial nitric oxide synthase (eNOS). This activation of eNOS (Ca2+/calmodulin-regulated) occurs through (i) the phosphorylation of NOS (phosphatidylinositol-3-OH-kinase–Akt mediated pathway), (ii) increased calcium flux induction, and (iii) recruitment of heat-shock protein-90. After activation, eNOS catalyzes the conversion of L-arginine to L-citrulline and NO (Figure 3). The produced NO stimulates neovascularization via angiogenesis as well as vasculogenesis. However, in PE, this protective signaling pathway of VEGF is compromised because of an increased circulating concentration of sFlt-1 and sEng and a decreased expression of PlGF. [33,42] An elevated concentration of circulating sEng and sFlt1 present in preeclamptic women may oppose the NO-dependent vasodilatation stimulated by VEGF, TGF-β, and PlGF, consequently leading to the development of hypertension observed in PE patients. Research reports that a decreased NO concentration significantly contributes to the pathogenesis of PE. The sFlt1- and sEng-induced inhibition of eNOS activation indicates the molecular basis for an increased mean arterial pressure [23]. Therefore, the attenuation of a VEGF-dependent activation of eNOS may inhibit angiogenesis and induce hypertension [23] (Figure 3).

In summary, elevated sFlt-1 concentration increases peripheral vascular resistance, which subsequently increases blood pressure. The increased sequestering of VEGF by sFlt-1 may also disturb the glomerular filtration barrier and cause glomerular endothelial injury, leading to proteinuria [44]. Furthermore, sFLT1 mRNA expression was also found to be increased in the placenta of women with PE. Studies on animal models suggest that administering exogenous sFlt-1 in experimental animals leads to glomerular endotheliosis, proteinuria, and hypertension [3].

### 3.5. Placental Protein 13 (PP-13)

PP-13 is another fetoplacental- specific molecule that is being used as a biomarker for predicting preeclampsia [25]. PP13 is amongst the 56 identified placental proteins described to date. PP13 is a carbohydrate binding protein (32 kDa homo-dimer protein, 139 amino acid residues) belonging to the galectin family and synthesized in the syncytiotrophoblast [12,45,46]. The structural and functional characteristics of PP-13 are vital in placental development and regulatory pathways [46]. It is involved in early placentation; however, it also plays a major role in the maintenance of pregnancy at different stages of gestation (viz., trophoblast invasion, maternal–fetal immune tolerance, embryo implantation, and vascular remodeling). The specificity of the conserved carbohydrate recognition domain for β-galactosides-containing glycoconjugates plays an important role in implantation and embryogenesis [46]. 

PP-13 can bind with the β-actin located in the trophoblastic cells, which enables their migration to the placental bed along with enhancing the secretion of prostacyclins required for spiral artery remodeling during early placentation. PP-13 also initiates apoptosis in maternal T cells for effective placentation and implantation. PP-13 also plays a significant role in trophoblast differentiation and syncytialization that helps in the secretion of immune proteins and placental hormones necessary for embryo development and immune tolerance. Research showed a decreased concentration of serum/plasma PP-13 during the first trimester of pregnancy; however, these concentrations gradually increase with the progress in the gestation period. Evidence suggests the presence of low serum concentration of PP-13 in PE [12,45,46].

It has been reported that in normal pregnant women, median serum PP-13 concentrations increase from 166 pg/mL to 202 pg/mL and 382 pg/mL in the first, second, and third trimesters, respectively [46]. Most of the research studies have reported that a reduced serum concentration of PP-13 during the first trimester increases the risk of developing PE. Research reveals that serum PP-13 concentrations in patients who developed early-onset PE (estimated during their first trimester) were significantly lower than those with normal gestation (specificity: 80%, sensitivity: 100%). A study conducted by Vasilache et al. aimed at using PP-13 for the prediction of PE and showed a specificity of 0.83 (95% CI) and a sensitivity of 0.53 (95% CI) [47]. Another study also reported that the maternal PP-13 mRNA expression was significantly reduced in PE patients (28%) compared to the levels observed in the control group (76%), with a highly statistically significant difference (P < 0.0001) [48]. These results suggest that serum PP-13 concentrations estimated during the first trimester may serve as a promising marker for the risk assessment of PE. Thus, estimating PP-13 concentration during the first trimester and using it as a screening marker for PE may help in the identification of women predisposed to develop early-onset PE [12,45,46].

Consequently, PP-13 may be considered a strong predictive factor in early-onset PE. Assessment of PP-13 (in the first trimester) in combination with uterine artery Doppler ultrasound may increase the prediction rate to 90% [12,25,46].

### 3.6. Growth Differentiation Factor 15 (GDF15)

GDF-15, a member of the TGF-β superfamily, is also known as a macrophage inhibiting cytokine-1 (MIC-1). It is produced in the placenta; however, it is also secreted in response to stress and is upregulated during cellular injury and inflammation. GDF-15 has also been recognized to possess a cardio-protective function [49,50].

Research has demonstrated that GDF-15 concentrations increase with gestational age and are dysregulated in PE. During the 30th–34th weeks of gestation, GDF-15 concentrations were found to be higher in women subsequently developing PE than in women with normal pregnancy; nevertheless, the difference was relatively minor. Furthermore, studies focused on assessing serum GDF-15 concentrations in women with PE have reported discrete findings with no change, decreased concentrations, and significantly increased concentrations. In the absence of consistent findings, the utility of GDF-15 concentrations (as an individual prediction biomarker) in clinical practice seems to be unlikely [49,50,51]. However, when used in combination with sFlt-1 and PlGF, GDF-15 may be a promising biomarker for the prediction of PE [50].

### 3.7. A disintegrin and Metalloprotease 12 (ADAM-12)

ADAM-12 is a multidomain glycoprotein derived from the placenta that possesses proteolytic and cell-adhesion activities. ADAM-12 controls the migration and invasion of trophoblasts during placental development. Hence, it is a key constituent in controlling the growth and development of the placenta and the fetus [52,53,54]. ADAM-12 occurs in the form of ADAM-12-L (long) and ADAM-12-S (short). ADAM-12-S, the secreted form of ADAM-12, possesses a proteolytic activity against IGFBP-3, which is believed to stimulate growth by promoting the IGF-I and IGF-II levels. ADAM-12-S is found in maternal serum beginning from the first trimester of pregnancy and increasing throughout gestation [52].

Studies report that the serum ADAM-12 concentrations in women predisposed to develop PE were significantly lower (P < 0.05) than those observed in women with normal pregnancy [25,52,54,55]. However, the available research indicates a modest predictive efficiency of ADAM-12 for PE [25,54,55]. Research suggests using a combination of screening methods for predicting PE. However, a combination of screening methods such as PAPP-A, β-hCG, PlGF, and ADAM-12 showed a detection rate of 44% only, with a false positivity rate of 5% [24].

### 3.8. β-Human Chorionic Gonadotropin (β-hCG)

hCG is a glycoprotein hormone with two non-covalently associated subunits, α and β. It is synthesized by placental trophoblasts. The free β-subunit can be either produced directly by trophoblast cells, become dissociated from hCG into free subunits (α and β), or become nicked by macrophages or neutrophils. The serum hCG concentration reaches a peak at 8–10 weeks of pregnancy, after which it declines and obtains a plateau at 18–20 weeks of pregnancy [56].

A reduced serum hCG concentration during early pregnancy may be an indication of an impaired invasion of trophoblast cells; consequently, a reduced serum hCG concentration may act as a biomarker for delayed implantation and impaired placental development. These factors may further contribute to the development of PE [57].

A study conducted by Asvold et al. showed that hCG concentrations were inversely associated with the risk of development of PE (dose-dependent). Women with hCG concentrations of <50 IU/l had a four times higher risk of developing severe PE compared to women with hCG concentrations ≥150 IU/l (Day 12 after transfer of cleavage stage embryos (2- to 4-cell stage) in pregnancies after IVF treatment) [57]. However, the study also reported that a single measurement of hCG concentration during early pregnancy may not serve as a potent biomarker for individual prediction of PE risk [57]. Other studies also demonstrate inconsistent findings. Few studies indicate that increased hCG and β-hCG concentrations during the second trimester of gestation were associated with a higher risk of PE [25,56,58], whereas another study revealed the absence of any statistically significant association between PE and β-hCG concentration [59]. However, all these studies suggest that the potential use of β-hCG as a biomarker for the prediction of PE shows a low detection rate with reduced sensitivity [25,57,58,60].

### 3.9. Inhibin Alpha (Inhibin-A)

Inhibin-A is a placenta-derived glycoprotein hormone belonging to the TGF-β superfamily. Inhibin-A is involved in trophoblast differentiation and proliferation, embryo implantation, and endometrial decidualization. It thus helps in fetal growth and maintenance of pregnancy. Inhibin-A concentration reaches its first peak at 8–10 weeks of gestation and becomes stable at 14–30 weeks of gestation, and later increases gradually during the third trimester and onwards, reaching its highest level at delivery [61,62].

Studies demonstrate that an increased Inhibin-A concentration during pregnancy is significantly associated with PE [61,62,63,64]. The possible reason of serum Inhibin-A levels becoming elevated in women with PE may be owing to the abnormal invasion and proliferation of trophoblasts in the uterine vessels in response to the repair of ischemic damage. The consequent damage and repair may lead to the functional changes on the surface of PE placenta, contributing to an increase in serum Inhibin-A concentration [61,65].

Studies suggest that Inhibin-A may be useful for the detection of PE [62,63,64]; however, the predictive sensitivity of Inhibin-A as a potent biomarker is relatively low, and so it is recommended to be used in combination with other biomarkers for best predictive outcome [25,62,63].

### 3.10. Soluble Endoglin (sEng)

sEng is another placenta-derived 65 kDa soluble form of the homodimeric transmembrane glycoprotein endoglin (Eng). It is an antiangiogenic factor that acts as a co-receptor for TGF-β1 and TGF-β3, and it is highly expressed in endothelial cells and trophoblasts. TGF-β is an anti-inflammatory growth factor. A prolonged exposure of endothelial cells to TGF-β stimulates the expression of the eNOS gene and protein [23]. sEng modulates TGF-β signalling by acting as an endogenous TGF-β1 inhibitor. An elevated concentration of sEng inhibits the TGF-β signalling pathway, eNOS activation and vasodilation, thereby interrupting significant homeostatic mechanisms essential for sustaining the vascular health [23,42]. Consequently, it antagonizes and impairs the biological action of the proangiogenic factor TGF-, an action like sFlt-1 antagonizing VEGF. This further indicates that sEng leads to impaired TGF-β signalling in the vasculature, thereby altering vascular permeability leading to hypertension [3,15,23,24,66].

Studies show that serum sEng concentration was upregulated in PE [15,23,24]. Venkatesha et al. reported that the elevated serum sEng concentration in PE individuals correlated with the severity of the disease. The sEng expression in PE was four times that of the normal pregnancy (P < 0.01). Furthermore, compared to controls (gestational age-matched), sEng concentrations were three, five, and ten times those in women with mild PE, severe PE, and HELLP syndrome, respectively [23]. Another study comparing the sEng concentration (ng/ml) in women with PE and normotensive pregnant women revealed a significantly higher sEng concentration in women with PE during the second trimester (MD: 5.554, P < 0.001) and the third trimester (MD: 31.006, P < 0.001). During the first trimester, the concentration of sEng was higher in women with PE; however, the difference was statistically non-significant (MD: 1.105, P = 0.06). Furthermore, the sEng concentrations were significantly higher in both early-onset and late-onset PE (P < 0.05) [55]. Furthermore, studies in an animal model showed that the effect of sEng was augmented by co-administration of sFlt1, causing severe PE with HELLP syndrome and restricted fetal growth [22].

An elevated sEng concentration may be observed before the onset of clinical symptoms, and its concentrations may be associated with the severity of the disease; measuring sEng concentration may allow an early prediction and diagnosis of PE [66,67,68].

The present review attempted to summarize the significant biomarkers that were discussed in the above section, but the major hinderance for drawing conclusions is the fact that the values of obtained biomarkers were collected from different countries. Most of the previous literature had limited or incomplete information regarding either PE or biomarkers. Hence, only a limited number of publications (a minimum of two papers to a maximum of four papers) had the information on PE onset with biomarker levels; the papers are represented in Table 1, providing the baseline characteristics of the denoted population. From Table 1, it can be observed that there was not much difference in the serum levels of maternal PAPP-A and PIGF levels documented between American and Chinese populations. The predominant angiogenic markers that were well demarcated through the literature survey were determined to be PAPP-A, PIGF, sFlt-1, and sEng. Most of the review discussed the usefulness of PAPP-A and PIGF, to a greater extent; in addition, for sFlt-1 and sEng, multicentric trials were performed in larger quantities. Hence, the baseline details of the substantial angiogenic biomarkers confined to PE are exhibited in Table 1 with country information. Table 2 provides details of combination biomarkers useful to PE. Since the data were limited in nature, the existing literature provided the baseline details of the sFlt-1:PIGF ratio in a clinical scenario through a multicentric study along with sensitivity and specificity.

Table 3 describes the summarized information of the control size, sample size, control value and sample value with respect to the examined biomarker through the literature assessed. This is presented to facilitate understanding of the broad geographical difference in the levels of significant biomarkers of PE existing between populations. It is obvious that PIGF levels were found to be 2.3 times lower in PE patients than the control group, and in our analysis, the study observed a greater time of fall in PIGF among German patients followed by the Indian, American and Chinese populations. Figure 3 illustrates the increasing trend of all represented biomarkers except PAPP-A with respect to the studied nation.

In general, the first-trimester combined algorithm for PE, which combines maternal characteristics with mean arterial blood pressure, mean uterine artery resistance, and circulating PlGF to stratify risk, was recently discussed in [69]. Clinical guidelines to apply a PE risk score through NICE and ACOG guidelines were also discussed. The study concluded that the sFlt1:PlGF ratio > 38 is a strong “rule out” test with a 99.3% negative predictive value for preeclampsia developing within a week. If the concentration of PlGF is 100 pg/mL, it represents a screen positive in women with suspected preeclampsia at or before 35 weeks, achieving a 96% sensitivity and a 98% negative predictive value for preeclampsia developing within two weeks. PIGF has been shown to reduce the time to diagnosis, adverse maternal outcomes, outpatient attendances and costs to healthcare service. 

Another significant study [70] showed that PAPP-A and PlGF MoMs (Multiples of Median) values were significantly reduced among early-onset PE cases (0.57 and 0.60), followed by preterm PE (0.63 and 0.67), all PE (0.74 and 0.74), and gestational hypertension (0.89 and 0.86) cases relative to controls (0.99 and 1.00) for first-trimester PAPP-A and PlGF, respectively. In addition, the study also displayed that a combination of maternal characteristics and PAPP-A and PlGF can provide reasonable performance for PE screening in the first trimester. In the second trimester, we found PlGF to be a better predictor for PE than the sFlt-1:PlGF ratio before 20 weeks of gestation.

**Table 1 ijms-24-13252-t001:** Global-wise reported concentration of maternal serum biomarker levels associated with early onset and late onset of PE corresponding to the trimester (gestational weeks).

Biomarker Name (Units)	Authors (Year)/ Country	Control (Healthy Population) Mean (Range)	EOP Mean (Range)	LOP Mean (Range)	Any Stage of PE	Trimester (GW)
PAPP-A (MoM)	Sonek et al. (2018) [71]/USA	1.00 (0.69–1.50)	0.62 (0.50–0.86)	0.97 (0.57–1.47)	NA	1st (11–13 GW)
	Hu et al. (2021) [72]/Chinese	1.01 (0.69–1.44)	0.70 (0.45–1.09)	0.86 (0.60–1.26) at term	NA	1st (11–13 GW)
PIGF (MoM)	Sonek et al. (2018) [71]/USA	1.01 (0.81–1.27)	0.68 (0.38–1.17)	1.07 (0.84–1.28)	NA	1st (11–13 GW)
	Hu et al. (2021) [72]/Chinese	0.99 (0.73–1.32)	0.91 (0.63–1.18)	0.83 (0.53–1.09)	NA	1st (11–13 GW)
sVEGFR-1 (pg/mL)	Kusanovic et al. (2009)/USA	1612 (245–10595.5)	NA	NA	1637.4 (325.1–17768.9)	NA
sFlt-1 (pg/mL)	Widmer et al. (2015) [73]/Multicentric (Argentina, Colombia, Peru, India, Italy, Kenya, Switzerland and Thailand)	2230 (1490–3340)	2030 (1300–2930)	NA	1890 (1210–2840)	1st–2nd (<20 GW)
		2280 (1480–3580)	2510 (1460–4310)	NA	2260 (1400–3650)	2nd (23–27 GW)
		3760 (2520–5800)	NA	NA	7905 (4750–13,620)	3rd (32–35 GW)
PP-13 (MoM)	Romera et al. ((2008) [74]/USA	1.00 (0.83–1.10)	0.26 (0.10–0.40)	0.24 (0.11–0.40)	0.59 (0.41–0.83)	NA
β-hCG (MoM)	Hanchard et al. (2020) [75]/Australia	2.27 ± 0.90	NA	NA	2.68 ± 1.10	NA
Inhibin-A (MoM)	Farzaneh et al. ((2019) [60]/NA	1.67 ± 0.59	NA	NA	1.98 ± 0.81	NA
sEng (pg/mL)	Widmer et al. (2015) [73]/Multicentric (Argentina, Colombia, Peru, India, Italy, Kenya, Switzerland and Thailand)	5.0 (3.9–6.4)	5.8 (4.3–8.4)	NA	5.5 (4.3–7.6)	1st–2nd (<20 GW)
		4.5 (3.5–5.6)	5.4 (4.2–7.1)	NA	5.5 (4.2–7.2)	2nd (23–27 GW)
		7.7 (5.6–11.0)	5.8 (4.3–8.4)	NA	17.4 (9.8–35.5)	3rd (32–35 GW)

**Note:** GW: Gestational Week; PE: Preeclampsia; EOP: Early-onset preeclampsia; LOP: Late-onset Preeclampsia.

**Table 2 ijms-24-13252-t002:** Comparison of predicting rates (%) for combinations of serum biomarkers for PE.

Biomarker in Combinations (Units)	Authors (Year)/ Country	Control (Healthy Population) Mean (Range)	EOP Mean (Range)	LOP Mean (Range)	Any Stage of PE	Trimester (GW)	Reliability Parameter
sFlt-1/PlGF	Widmer et al. (2015) [73] /Multicentric (Argentina, Colombia, Peru, India, Italy, Kenya, Switzerland and Thailand)	27.9 (12.7–62.2)	30.7 (14.7–81.7)	NA	32.8 (16.1–80.9)	1st–2nd (<20 GW)	Cut-off <117 Specificity: 90% Sensitivity: 10%

**Table 3 ijms-24-13252-t003:** Country-wise distribution of average value of the significant biomarkers for detecting PE.

Biomarker Name	Country/no of Literature Assessed	Authors (Year)	Control Size (Mean ± SD)	PE sample Size (Mean ± SD)	Control Value (Mean ± SD)	PE Sample Value (Mean ± SD)	Ratio of Control to the PE
PIGF (pg/mL)	USA/n = 3	Levine et al. (2004); Holston. (2009); Young et al. (2009) [76,77,78]	629.3 ± 814.7	21.7 ± 15.9	561.3 ± 155.2	180.8 ± 83.2	3.1
	Germany/n = 2	Schmidt et al. (2009); Engels et al. (2013) [79]	114 ± 99.0	35.5 ± 40.3	288.4 ± 293.9	55.2 ± 17.7	5.2
	China/n = 3	Ouyang et al. (2009); Ding et al. (2018); Wang et al. (2021) [80,81,82]	182 ± 153.1	89.3 ± 43.8	263 ± 33.3	132.4 ± 103.8	1.2
	India/n = 2	Aggarwal et al. (2012); Kumar et al. (2023) [83,84]	77.5 ± 43.1	75.5 ± 48.8	368.8 ± 182.2	86.2 ± 14.3	4.3
sFlt1 (pg/mL)	USA/n = 4	[76,77,78,85]	482.5 ± 727.1	26.8 ± 16.5	4978.5 ± 3831.3	12460 ± 8764.3	-
	China/n = 3	Ouyang et al. (2009); Ding et al. (2018); Wang et al. (2021) [80,81,82]	182.3 ± 153.1	89.3 ± 43.8	2041.2 ± 464.9	4524.4 ± 2428.3	-
	India/n = 2	Aggarwal et al. (2012); Kumar et al. (2023) [83,84]	77.5 ± 43.1	75.5 ± 48.8	5704.5 ± 573.5	27650 ± 19586.7	-
sFlt1/PIGF	USA /n = 2	Levine et al. (2004); Young et al. (2009) [76,78]	802 ± 1077.6	25.5 ± 20.5	16.5 ± 16.3	183 ± 145.7	-
	China/n = 2	Ouyang et al. (2009); Wang et al. (2021) [80,82]	98.5 ± 68.6	66.0 ± 24.0	6.7 ± 0.14	16.7 ± 4.8	-
	India/n = 2	Aggarwal et al. (2012); Kumar et al. (2023) [83,84]	77.5 ± 43.1	75.5 ± 48.8	147.9 ± 181.7	670.8 ± 112.7	-
sEng	USA/n = 3	Levine et al. (2004); Holston., (2009); Young et al. (2009) [76,77,78] [76,77,78]	619.3 ± 825.1	30.3 ± 16.7	9.9 ± 1.3	36.1 ± 15.7	-
	Chinese/n = 3	Wang et al. (2021); Fang et al. (2010); Zhang et al. (2020) [82,86,87]	27.7 ± 12.5	46.3 ± 7.1	10.2 ± 1.1	25.5 ± 11.9	-
	India/n = 2	Sachan et al. (2016); Archana et al. (2018) [77]	35.0 ± 7.1	32.5 ± 3.5	4.0 ± 2.7	11.9 ± 4.5	-

## 4. Conclusions

The existing comprehensive research suggests that, from a variety of biomarkers that can be applied in clinical settings to diagnose PE, three biomarkers were found to be sensitive either to detect or rule out the condition, including PlGF, PAPP-A and PlGF and sFlt1:PlGF.

## 5. Future Perspectives

Consequently, larger prospective studies focused on screening the best predictive biomarkers with better predictive values (when used alone or in combination) should be carried out. Moreover, research to completely understand the molecular mechanism of these biomarkers in the development of PE should also be conducted. Such a specific prediction strategy for detecting PE in early gestation may help in identifying high-risk women (predisposed to develop PE); accordingly, a specific preventative intervention/therapy may be provided. Furthermore, an early diagnosis may also alleviate anxiety and redundant therapy/interventions in women with low risk of developing PE. Perhaps it is worth asking whether these markers will not only serve to predict early PE, but also, in the future, serve to differentiate between the early and late forms, since we know that the late form of PE only in a fraction of cases runs with a disruption of these markers. Further studies in the future will help answer this question.

## Figures and Tables

**Figure 1 ijms-24-13252-f001:**
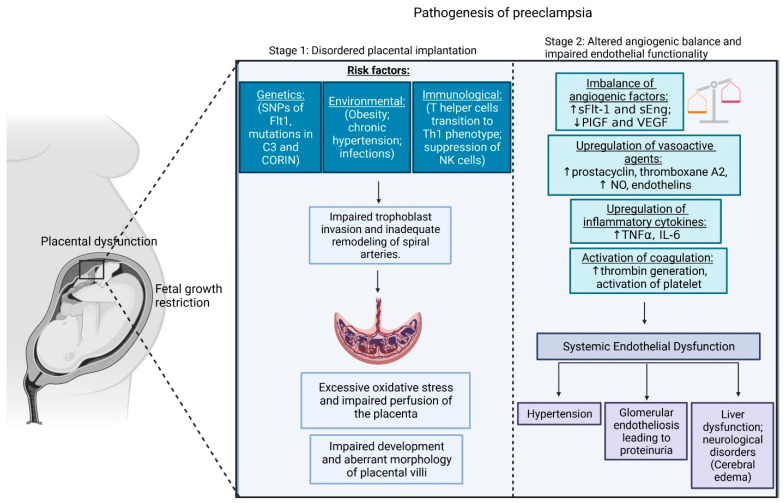
Two stages of preeclampsia pathogenesis. NOTE: Preeclampsia has a two-stage pathogenesis. Preclinical Stage 1 is characterized by abnormal placentation, resulting in the emission of soluble factors into the maternal blood, which then causes systemic endothelial dysfunction and hypertension (Stage 2). Created with BioRender.com.

**Figure 2 ijms-24-13252-f002:**
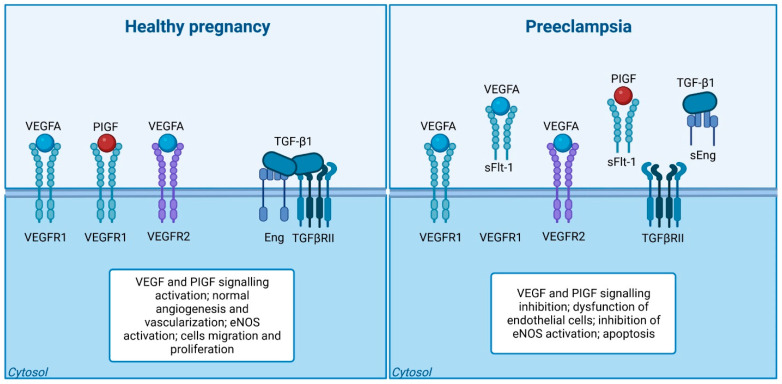
The Disrupted Balance: Angiogenic Imbalance in Preeclampsia. NOTE: Imbalance of angiogenesis in preeclampsia. In a normal pregnancy, the balance and stability of blood vessels are regulated by appropriate levels of vascular endothelial growth factor (VEGF) and transforming growth factor-1 (TGF-β1) signaling. However, in the case of preeclampsia, there is an excessive release of two antiangiogenic proteins, sFlt-1 and sEng, by the placenta. These proteins act to inhibit the signaling of VEGF and TGF-β1 in the blood vessels. Consequently, this disruption leads to the dysfunction of endothelial cells, characterized by a decrease in the production of nitric oxide. Created with BioRender.com.

**Figure 3 ijms-24-13252-f003:**
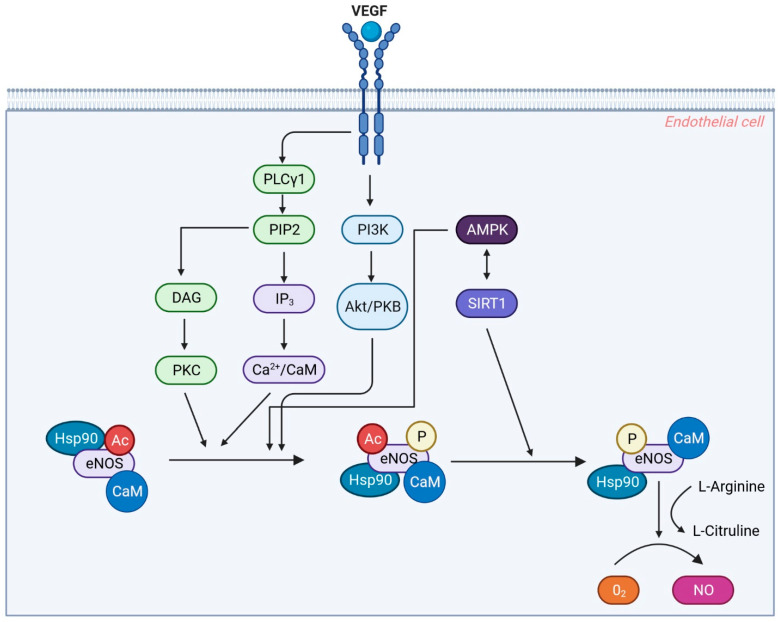
Regulation of Endothelial Nitric Oxide Synthase (eNOS) by VEGF Signaling Pathways. NOTE: Regulation of endothelial nitric oxide synthase (eNOS). Upon binding of VEGF to its receptor (VEGFR2), the receptor dimerizes and activates its tyrosine kinase activity, leading to autophosphorylation of intracellular domains. This event triggers a series of signaling pathways that modulate NO synthesis. Firstly, VEGFR2 activation stimulates the PI3K/Akt pathway, resulting in an increase in intracellular Ca^2+^ levels, which induce the binding of calmodulin (CaM) to endothelial nitric oxide synthase (eNOS), facilitating its activation. Additionally, VEGFR2 signaling activates PLCγ, leading to the conversion of PIP2 into DAG and IP3. IP3 acts as a secondary messenger contributing to the elevation of intracellular Ca^2+^ levels. On the other hand, DAG activates PKC, which plays a role in downstream signaling events. Hsp90, a molecular chaperone, is recruited to the activated VEGFR2 complex and assists in the proper folding and stabilization of eNOS, ensuring its functional integrity and preventing degradation. The coordinated actions of Ca^2+^, calmodulin, and Hsp90 lead to the activation of eNOS, enabling the conversion of molecular oxygen and L-arginine to produce NO. The generated NO mediates various biological effects, including enhanced vascular permeability, vasorelaxation, and maintenance of endothelial cell survival. Created with BioRender.com.

## Data Availability

Not applicable.

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
