# Peer review of "From Biomarkers to the Molecular Mechanism of Preeclampsia—A Comprehensive Literature Review"

_ijms, 2023, doi:10.3390/ijms241713252_

Round 1
Reviewer 1 Report
This is a literature review of biomarkers in pre-eclampsia. Overall it describes pre-eclampsia pathophysiology and biochemistry of biomarkers well, but is lacking in clinical relevance and novel developments in the field
Major comments:
The authors state 'no preventative strategies are available' at various points in manuscript e.g. line 58 - this is not the case. Aspirin has been shown to prevent pre-eclampsia (especially preterm pre-eclampsia) in high risk women. This is the only reason screening is of interest - because we have an intervention that works.
The clinical relevance of these biomarkers is missing. Whilst the biochemistry is well described, the current description suggests all the biomarkers described have similar potential, which is misleading. A lot of work has been done demonstrating the utility of PlGF as a rule in/out test for pre-eclampsia <37 weeks in clinical practice (see PARROT trial: https://www.thelancet.com/journals/lancet/article/PIIS0140-6736(18)33212-4/fulltext, updated NICE guidance in UK: https://www.nice.org.uk/guidance/dg49). s-Flt has also entered clinical practice in some centres. This needs to be clearer - we are already using PlGF routinely in the UK, and it is also used in the FMF algorithm for 1st trimester pre-eclampsia screening, not B-hcg or any of the others suggested here.
Finally please see new Nature Medicine Paper for novel potential biomarkers of pathophysiology: https://www.nature.com/articles/s41591-023-02374-9, the paper feels lacking in novelty and there some interesting pathways to suggest further investigation here.
Finally I am unaware of strong evidence of complement or CORA SNPs in pathophysiology of pre-eclampsia - see GWAS above.
Some editing required but of good standard.
Author Response
Dear Reviewer,
thank you very much for the revision of the manuscript, and your valuable comments and suggestions. We have edited the manuscript, please see the point-by-point answers:
The authors state 'no preventative strategies are available' at various points in manuscript e.g. line 58 - this is not the case. Aspirin has been shown to prevent pre-eclampsia (especially preterm pre-eclampsia) in high risk women. This is the only reason screening is of interest - because we have an intervention that works.
The authors state that , there is no consensus on the etiology of this disease, and neither is their preventive strategy, We have edited manuscript to clarification.
The clinical relevance of these biomarkers is missing. Whilst the biochemistry is well described, the current description suggests all the biomarkers described have similar potential, which is misleading. A lot of work has been done demonstrating the utility of PlGF as a rule in/out test for pre-eclampsia <37 weeks in clinical practice (see PARROT trial: https://www.thelancet.com/journals/lancet/article/PIIS0140-6736(18)33212-4/fulltext, updated NICE guidance in UK: https://www.nice.org.uk/guidance/dg49). s-Flt has also entered clinical practice in some centres. This needs to be clearer - we are already using PlGF routinely in the UK, and it is also used in the FMF algorithm for 1st trimester pre-eclampsia screening, not B-hcg or any of the others suggested here.
We have added Table 2 and Table 3 with an analysis of the clinical relevance of biomarkers.
Finally please see new Nature Medicine Paper for novel potential biomarkers of pathophysiology: https://www.nature.com/articles/s41591-023-02374-9, the paper feels lacking in novelty and there some interesting pathways to suggest further investigation here.
This is very interesting paper, but we focused on clinical useful markers in this paper.
Finally I am unaware of strong evidence of complement or CORA SNPs in pathophysiology of pre-eclampsia - see GWAS above.
We agree that there are still lack of consensus of pathophysiology of pre-eclampsia.
Best wishes
Magda Rybak-Krzyszkowska
Reviewer 2 Report
This manuscript describes molecular mechanism and biomarkers of preeclampsia.
It only requires minor revision as listed below.
1. Introduction
L100 Early onset and late onset do not imply stage 1 and stage 2, respectively. Stage 1 and Stage 2 should be deleted.
Author Response
Dear Reviewer,
thank you very much for the revision of the manuscript. We have edited the introduction as suggested.
Best wishes
Magda Rybak-Krzyszkowska
Reviewer 3 Report
The manuscript contains interesting perspectives for prediction of preeclampsia using serum biomarkers. The authors showed good knowledge of the subject and provide enough information about existing biomarkers which can be used in clinical practice. The manuscript is generally well written. However, I have a few concerns which are listed below for the authors to improve their article.
1. The manuscript is poorly structured and hard for understanding. The text is sometimes chaotic and contains a lot of repeats. For example:
- Section 2: I would suggest to indicate the features of early and late PE pathogenesis in different sub-sections corresponding to Figure 1.
- Sub-section 3.3: the appearance of NO in sub-section focused on sFlt-1 is not clear. Probably, eNOS pathway should be put into separate sub-section.
2. The work would benefit from adding the Table summarizing the concentrations of described biomarkers (PlGF, VEGF, sFlt, etc.) in all trimesters of normal pregnancy and in early and late (mild and severe) PE.
3. The conclusions on validity of biomarkers models for prediction and diagnosis of PE are not sufficient. The recommendations are not presented. So, the second Table could contain the comparison of predicting rates (%) for combinations of serum biomarkers proposed to date in order to reveal the most reliable ones.
The Einglish is generally fine.
Author Response
Dear Reviewer,
thank you very much for the revision of the manuscript, and your valuable comments and suggestions. We have edited the manuscript, please see the point-by-point answers:
- The manuscript is poorly structured and hard to understand. The text is sometimes chaotic and contains a lot of repeats.
The Authors also do agree with the reviewer’s comments; hence we have edited the text currently to our best capabilities to avoid repeats.
- Section 2: I would suggest indicating the features of early and late PE pathogenesis in different sub-sections corresponding to Figure 1
Yes, as per the reviewer’s suggestion, the authors have made the following corrections.
- Novel line “Sufficient blood flow to the placenta is essential for a pregnancy to be effective” was added in line number 95 (edited document)
- The following line “During normal implantation, the placental trophoblasts (of fetal origin) invade the uterus inducing the remodeling of uterine spiral arteries, making the arteries capable of accommodating increased blood flow, thereby providing adequate placental perfusion to nourish the growing fetus.” from line number 110 to 113 of the original document was removed and inserted to line number inserted from line number 95 (Edited document.
- “The placenta is a major contributing factor to the pathogenesis of PE” was removed from line number 95 of the original document and inserted to line number 99 of the edited document.
- “The pathogenesis of PE shows significant endothelial damage which is induced by factors such as immune system aberrations, increased oxidative stress, hypoxia, and genetic susceptibility [such as mutations in genes encoding complement component 3, or mutations in corin (a cardiac protein)]. These factors may cause placental dysfunction that further leads to the release of antiangiogenic factors [viz. soluble fms-like tyrosine kinase 1 (sFlt1) and soluble endoglin (sEng)], and inflammatory mediators inducing hypertension, proteinuria, and other complications observed in PE (Figure 1). [14]–[17]” line number 103 to 109 from the original document was removed and inserted line number 104 to 111 of the edited document.
- “PE may be classified as early-onset (stage 1) and late-onset (stage 2) depending on the timing, pathophysiology and clinical implications.” Was removed from line number 100 to 101 (original document) and added to the line number 112 to 113 (Edited document).
- “In PE, this spiral artery remodelling is impaired (due to the failure of the trophoblasts to adopt an endothelial phenotype and its adhesion molecules), and owing to acute atherosis there is reduced blood flow to the placenta. This impairment may cause the placenta to get deprived of oxygen, causing ischemia and increased oxidative stress. Research demonstrates that severe PE has been associated with placental hypoperfusion and ischemia. [15], [17]” line number 113 to 118 was removed (Original document)
- “The condition may be categorized as early-onset preeclampsia, requiring delivery prior to 34 weeks of gestation based on time.” line was added in line number 116 to 117 below (adding) a section heading 2.1. Early-onset PE (line number 115 of the edited document)
- “Late-onset PE” Section was made in line number 135. “Preeclampsia detected at or after 34 weeks of pregnancy is referred to as having a late onset” line number 136 to 137 (Edited document) was added.
- Further on lines were adjusted to get the flow in the sections.
- Sub-section 3.3: the appearance of NO in the sub-section focused on sFlt-1 is not clear. Probably, the eNOS pathway should be put into a separate sub-section.
“3.4. Endothelium-derived Nitric oxide (NO)” section was inserted in line number 310 (edited document)
- The work would benefit from adding the Table summarizing the concentrations of described biomarkers (PlGF, VEGF, sFlt, etc.) in all trimesters of normal pregnancy and in early and late (mild and severe) PE.
No. The authors disagree with the reviewer’s comment on “making a Table summarizing the concentration of described biomarkers in normal pregnancy PE” as the present study is not a meta-analysis review.
- The conclusions on the validity of biomarkers models for the prediction and diagnosis of PE are not sufficient. The recommendations are not presented. So, the second Table could contain the comparison of predicting rates (%) for combinations of serum biomarkers proposed to date to reveal the most reliable ones.
Conclusion section is only providing the idea of possible biomarkers proposing suggestion as the present study didnot followed inclusion and exclusion crtieria to comment on reviewer’s for assessing “prediction and diagnosis of PE, and prediction rates identify the reliable marker” since the present work is only a comprehensive review and not a meta-analysis review to evaluate the prediction rates to identify the reliable marker.
Reviewer 4 Report
The article is devoted to the actual topic of the search for predictors of preeclampsia. Despite numerous studies, the pathogenesis of preeclampsia is poorly understood. There is also no targeted therapy for preeclampsia. However, the authors concentrated on markers that have been known for more than 10 years. They are described in detail in the review. Visual graphic schemes are given. However, this review is not suitable for a meta-analysis. Each country has different criteria for preeclampsia. And that's a big problem in evaluating gene and protein expression data. Maybe reflect this in the article. The difference in approaches to this problem in different countries, or at least over the past 10 years. If the authors are considering well-known markers, it would be better if all submitted papers met the criteria for preeclampsia, such as EBCOG, because the criteria for preeclampsia (arterial blood pressure level, proteinuria in each country is different).
This problem is especially true for Asian countries. For example, in China there was a different classification according to the severity of preeclampsia. Although a lot of articles and data are published.
And this is a problem, of course, in the evaluation of the data. The list of references should be corrected. For example, reference 5. In general, I propose that the review should preferably be made more debatable with new modern accents.
The English language is not native for the authors, in view of this, the English language should be revised.
Author Response
Dear Reviewer,
Thank you for the revision of our manuscript and your valuable comments and suggestions. We have edited the manuscript, please see point-by-point answer below:
- The article is devoted to the actual topic of the search for predictors of preeclampsia.Despite numerous studies, the pathogenesis of pre-eclampsia is poorly understood. There is also no targeted therapy for pre-eclampsia. However, the authors concentrated on markers that have been known for more than 10 years. They are described in detail in the review. Visual graphic schemes are given. However, this review is not suitable for a meta-analysis. Each country has different criteria for preeclampsia. And that's a big problem in evaluating gene and protein expression data. Maybe reflect this in the article. The difference in approaches to this problem in different countries, or at least over the past 10 years. If the authors are considering well-known markers, it would be better if all submitted papers met the criteria for preeclampsia, such as EBCOG, because the criteria for preeclampsia (arterial blood pressure level, proteinuria in each country is different).
Yes. We do agree with the reviewer’s opinion that the current manuscript is not a meta-analysis, it is only a comprehensive literature review. Hence, there are inclusion and exclusion criteria in the present study to exactly comment on the most reliable and ideal biomarker for PE.
- And this is a problem, of course, in the evaluation of the data.
We have neither did a systematic research or meta-analysis, therefore data evaluation could not be performed.
- The list of references should be corrected.For example, reference 5.
Will be improvised based on reviewers’ comments regarding references.
Best regards,
Magda Rybak-Krzyszkowska
Round 2
Reviewer 3 Report
I have no comments.
Author Response
Thank you very much.
Reviewer 4 Report
The main remark is the almost complete lack of new data in this article. The changes made have improved the presentation of the article. The authors have systematized known data regarding preeclampsia. If the Editorial Board considers it possible to publish this review, it may be useful for graduate students, obstetricians and gynecologists in their beginning work.
Minor editing of English language required
Author Response
Dear Reviewer,
We would like to thank the Editor and the expert Reviewers again for their invaluable evaluation of our manuscript entitled: From Biomarkers to the Molecular Mechanism of Pre-Eclampsia – A Comprehensive Literature Review”
The article was again corrected by our team to improve its value. All changes in the manuscript are marked up using the “Track Changes”. However, if the Expert Consultant Editor would like to add the new subchapters on the potential, still under research consideration molecular markers e.g. free cell RNA, long non-coding RNA, or miRNA, we can add that information. We do not know whether the supplementation of our article at this steep review process would be possible, therefore we send the request to the Editor.
We hope that the value of the revised version of the manuscript is good and the article will be accepted for publication in the International Journal of Molecular Science.
Sincerely yours,
Magda Rybak-Krzyszkowska